# EcoFormer: Energy-Saving Attention with Linear Complexity

**Jing Liu**[*]   **Zizheng Pan**[*]   **Haoyu He**   **Jianfei Cai**   **Bohan Zhuang**[†]

Department of Data Science & AI, Monash University, Australia

## Abstract

Transformer is a transformative framework for deep learning which models sequential data and has achieved remarkable performance on a wide range of tasks, but with high computational and energy cost. To improve its efficiency, a popular choice is to compress the models via binarization which constrains the floating-point values into binary ones to save resource consumption owing to cheap bitwise operations significantly. However, existing binarization methods only aim at minimizing the information loss for the input distribution statistically, while ignoring the pairwise similarity modeling at the core of the attention mechanism. To this end, we propose a new binarization paradigm customized to high-dimensional softmax attention via kernelized hashing, called EcoFormer, to map the original queries and keys into low-dimensional binary codes in Hamming space. The kernelized hash functions are learned to match the ground-truth similarity relations extracted from the attention map in a self-supervised way. Based on the equivalence between the inner product of binary codes and the Hamming distance as well as the associative property of matrix multiplication, we can approximate the attention in linear complexity by expressing it as a dot-product of binary codes. Moreover, the compact binary representations of queries and keys in EcoFormer enable us to replace most of the expensive multiply-accumulate operations in attention with simple accumulations to save considerable on-chip energy footprint on edge devices. Extensive experiments on both vision and language tasks show that EcoFormer consistently achieves comparable performance with standard attentions while consuming much fewer resources. For example, based on PVTv2-B0 and ImageNet-1K, EcoFormer achieves a 73% reduction in on-chip energy footprint with only a slight performance drop of 0.33% compared to the standard attention. Code is available at https://github.com/ziplab/EcoFormer.

## 1   Introduction

Recently, Transformers [57] have shown rapid and exciting progress in natural language processing (NLP) [15, 14] and computer vision (CV) [17, 56] due to its extraordinary representational power. Compared with convolutional neural networks (CNNs) [30], Transformer models are generally more scalable to massive amounts of data and better at capturing long-dependency global information with less inductive bias, thus achieving better performance in many tasks [26, 38]. However, the efficiency bottlenecks, especially the high energy consumption, greatly hamper the massive deployment of Transformer models to resource-constrained edge devices, such as mobile phones and unmanned aerial vehicles, for solving a variety of real-world applications.

---

[*]Authors contributed equally.
[†]Corresponding author. Email: bohan.zhuang@monash.edu

36th Conference on Neural Information Processing Systems (NeurIPS 2022).

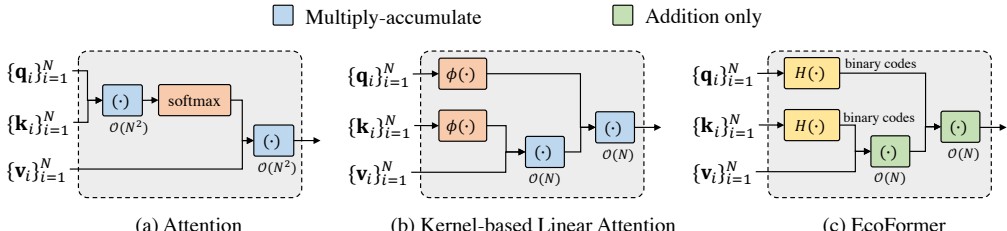

Figure 1: Computational graphs for standard attention (left), kernel-based linear attention (middle) and the proposed EcoFormer based on kernelized hashing (right).

Table 1: Energy cost for different operations (on 45nm CMOS technology) [21, 22, 32].

| Operation | 16-bit FP Add | 16-bit FP Mult | 32-bit FP Add | 32-bit FP Mult |
|---|---|---|---|---|
| Energy (pJ) | 0.4 | 1.1 | 0.9 | 3.7 |
| Area ($\mu m^2$) | 1,360 | 1,640 | 4,184 | 7,700 |

To reduce the energy consumption, quantization has been actively studied to lower the bit-width representation of network weights [11, 52, 66] and/or activations [25, 68, 67]. With the most aggressive bit-width, binary quantization [49, 2, 46] has attracted much attention since it enables efficient bit-wise operations by representing values with a single bit (*e.g.*, +1 or -1). When we only binarize weights in analogous to BinaryConnect [11], as shown in Table 1, it brings great benefits to dedicated hardware by replacing a large number of energy-hungry multiply-accumulate operations with simple energy-efficient accumulations, which saves significant on-chip area and energy required to run inference with Transformers, making them feasible to be deployed on mobile platforms with limited resources. However, the conventional binarization process typically targets at minimizing quantization errors between the original full-precision data distribution and the quantized Bernoulli distribution statistically. In other words, each token is binarized separately, where the binary representations may not well preserve the original similarity relations among tokens. This motivates us to customize the binarization process to softmax attention, the core mechanism in Transformer that encodes the pairwise similarity between tokens. To this end, we can adapt the well-established hashing methods, to map the high-dimensional queries and keys into compact binary codes (*e.g.*, 16-bits) that are able to preserve the similarity relations in Hamming space. A simple solution is to use the locality-sensitive hashing (LSH) [1] to substitute the binary quantization counterparts.

Nevertheless, another energy bottleneck exists in Transformers. Specially, given a sequence of tokens, the softmax attention obtains the attention weights by computing the inner product between a query token and all key tokens, leading to the quadratic time complexity $\mathcal{O}\left(N^2\right)$ regarding the number of tokens $N$, as shown in Figure 1 (a). This problem is even worse for a long sequence length $N$, especially for high resolution images in dense prediction tasks. To reduce the complexity of the softmax attention, some prior works propose to express the attention as a linear dot product of kernelized feature embeddings [45, 9]. With the associative property of matrix multiplication, the attention operation can be approximated in linear complexity $\mathcal{O}(N)$, as illustrated by Figure 1 (b).

Based on the hashing mechanism and kernel-based formulation of attention, we devise a simple yet effective energy-saving attention, called EcoFormer, which is shown in Figure 1 (c). In particular, we propose to use kernelized hashing with RBF kernel to map the queries and keys to compact binary codes. The resulting codes are valid for similarity preserving based on the good property that the codes' inner product (*i.e.*, Hamming affinity) and Hamming distance have one-to-one correspondence [36]. Thanks to the associative property of the linear dot-product between the binary codes, the kernelized hashing attention is in *linear complexity* with significant energy saving. Moreover, the pairwise similarity matrix in attention can be directly used to obtain the supervision labels for hash function learning, delivering a novel *self-supervised* hashing paradigm. By maximizing the Hamming affinity on the similar pairs of tokens and simultaneously minimizing on the dissimilar pairs of tokens, the pairwise similarity relations between tokens can be preserved. With *low-dimensional binary* queries and keys, we can replace most of the energy-hungry floating-point multiplications in attention with simple additions, which greatly saves the on-chip energy footprint.

To sum up, we make three main contributions: 1) We propose a new binarization paradigm to better preserve the pairwise similarity in softmax attention. In particular, we present EcoFormer, an energy-saving attention with linear complexity powered by kernelized hashing to map the queries and keys into compact binary codes. 2) We learn the kernelized hash functions based on the ground-

truth Hamming affinity extracted from the attention scores in a self-supervised way. 3) Extensive experiments on CIFAR-100, ImageNet-1K and Long Range Arena show that EcoFormer is able to significantly reduce the on-chip energy cost while preserving the accuracy. For example, based on PVTv2-B0 and ImageNet-1K, EcoFormer achieves a 73% reduction in on-chip energy footprint with only a marginal performance drop of 0.33% compared to the standard attention.

## 2 Related Work

**Efficient attention mechanisms.** To alleviate the quadratic computational cost for vanilla attention with respect to the number of tokens, much work has endeavored developing efficient attentions. One line of research performs attention only on the part of the tokens [38, 61, 4, 29, 60, 12, 58, 54]. For instance, Reformer [29], SMYRF [12], Fast Transformers [58] and LHA [54] restrict the attention to the most similar token pairs via hashing and reduces the computational complexity to $\mathcal{O}(N \log N)$. Linformer [60] approximates the attention with low-rank factorization that reduces the length of the key and value. However, the computational complexity is dependent on the design for reducing tokens. Another line of research speeds up the vanilla attention with kernel-based methods [9, 64, 41, 28, 47]. For example, Performer [9] approximates the softmax operation with orthogonal random features. Nyströmformer [64] and SOFT [41] approximate the full self-attention matrix via matrix decomposition. Although impressive achievements have been achieved, how to develop attention that is highly energy-efficient remains under-explored, as multiplications dominate the on-chip energy consumption. AdderNet variants [8, 53] replace the energy-hungry multiplication-based similarity measurement with the energy-efficient addition-based L1 distance and argue that additions can also provide powerful feature representations. Nevertheless, this heuristic approach brings a drastic performance drop. In contrast, we are from the perspective of binarization and propose to learn kernel-based hash functions using attention scores to map the original features into compact similarity-preserving binary codes in Hamming distance, which is energy-efficient and in linear complexity $\mathcal{O}(N)$. With the low-dimensional binary queries and keys, our EcoFormer is able to replace most of the multiplications with simple accumulations.

**Hashing.** Hashing is an efficient nearest neighbor search method by embedding the high-dimensional data into a similarity preserving low-dimensional binary codes, based on the intuition that highly similar data should be assigned the same hash key. Hashing methods can be roughly categorized into data-independent and data-dependent schemes. The former focuses on building random hash functions and locally sensitive hashing (LSH) [19, 7] is arguably the most representative one, which guarantees the sub-linear time similarity search and is followed by non-linear extensions such as hashing with kernels [31] or on manifolds [63]. The latter can be further divided into unsupervised [37, 20, 35], semi-supervised [44, 59] and supervised hashing [36, 71, 65]. When it comes to Transformers, as self-attention encodes the pairwise similarity among tokens, hashing is thus a natural choice to efficiently retrieve similar keys given a query. Reformer is such a pioneering work, which proposes to group similar tokens in a single hash bucket to form sparse self-attention. Our EcoFormer is fundamentally different from Reformer in three aspects: 1) Reformer relies on local attention lookups to reduce the complexity while our EcoFormer is designed from a numerical perspective, where the low-dimensional binary codes are used to save the multiplications; 2) Reformer is built upon LSH with linear mapping, which cannot deal with the kernel-based formulation of attention to scale linearly with the sequence length; 3) Our hash functions are self-supervised by the pairwise affinity labels in attention, which are optimized in conjunction with network parameters and more accurate than unsupervised random projections.

**Binary quantization.** Binarization, an extreme quantization scheme, seeks to represent the vectors by binary codes. As a result, the computationally heavy matrix multiplications become light-weight bitwise operations (*i.e.*, xnor and popcount), yielding promising memory saving and acceleration. In general, to make binary neural networks [24] reliable in accuracy, current research targets at tackling two main challenges. The first challenge is to minimize the quantization error, basically based on learning the scaling factors [49, 6], parameterizing the quantization range and/or intervals [27, 18], and ensembling multiple binary bases [34, 72], *etc*. Another category of studies focus on solving the non-differentiable optimization problem due to the discretization process via training with regularization [16], knowledge distillation [43, 46], relaxed optimization [23, 3], appending full-precision branches [39, 42] and so on. Apart from CNNs, there are some recent pioneering attempts targeting on binarizing Transformers. For example, BinaryBERT [2] proposes to push Transformer quantization to the limit by weight binarization. BiBERT [46] quantizes both weights,

embeddings and activations of BERT [15] to 1-bit and achieves considerable savings on FLOPs and model size, but still has obvious performance drop. In contrast, we propose to customize the binarization paradigm to softmax attention from the hashing perspective, preserving high-fidelity pairwise similarity information in compact binary codes which are used to deliver linear-complexity, energy-efficient self-attention.

## 3 Preliminaries

### 3.1 Attention Mechanism

Let $\mathbf{X} \in \mathbb{R}^{N \times D}$ be the input sequence into a standard multi-head self-attention (MSA) layer, where $N$ is the length of the input sequence and $D$ is the number of hidden dimensions. A standard MSA layer calculates a sequence of query, key and value vectors with three learnable projection matrices $\mathbf{W}_q, \mathbf{W}_k, \mathbf{W}_v \in \mathbb{R}^{D \times D_p}$, which can be formulated as

$$\{\mathbf{q}_t\}_{t=1}^N = \mathbf{X}\mathbf{W}_q, \{\mathbf{k}_t\}_{t=1}^N = \mathbf{X}\mathbf{W}_k, \{\mathbf{v}_t\}_{t=1}^N = \mathbf{X}\mathbf{W}_v, \tag{1}$$

where $D_p$ refers to the number of dimensions for each head. For each query vector, the attention output is a weighted-sum over all value vectors as

$$\text{Attention}(\mathbf{q}_t, \{\mathbf{k}_i\}, \{\mathbf{v}_i\}) = \sum_i \frac{\exp(\mathbf{q}_t \cdot \mathbf{k}_i / \tau)}{\sum_j \exp(\mathbf{q}_t \cdot \mathbf{k}_j / \tau)} \mathbf{v}_i, \tag{2}$$

where $\tau$ is the temperature for controlling the flatness of softmax and $\exp(\langle \cdot, \cdot \rangle)$ is an exponential function. With $N$ token, the computation of attention has a quadratic complexity of $O(N^2)$ in both space and time, which results in huge computational cost when dealing with long sequences.

### 3.2 Kernel-based Linear Attention

The idea behind kernel-based linear attention is to express the similarity measure in Eq. (2) as a linear dot-product of kernel embeddings, such as polynomial kernel, exponential or RBF kernel. A particular choice is to employ the finite random mapping [48] $\phi(\cdot)$ to approximate the infinite dimensional RBF kernel. Then, according to the theorem from Rahimi [48], the inner product between a pair of transformed vectors $\mathbf{x}$ and $\mathbf{y}$ with $\phi(\cdot)$ can approximate a Guassian RBF kernel. This gives rise to an unbiased estimation to $\exp(\langle \cdot, \cdot \rangle)$ in Eq. (2), which can be expressed as

$$\begin{aligned} \exp\left(\mathbf{x} \cdot \mathbf{y} / \sigma^2\right) &= \exp\left(\|\mathbf{x}\|^2 / 2\sigma^2 + \|\mathbf{y}\|^2 / 2\sigma^2\right) \exp\left(-\|\mathbf{x} - \mathbf{y}\|^2 / 2\sigma^2\right) \\ &\approx \exp\left(\|\mathbf{x}\|^2 / 2\sigma^2 + \|\mathbf{y}\|^2 / 2\sigma^2\right) \phi\left(\mathbf{x}\right)^\top \phi\left(\mathbf{y}\right). \end{aligned} \tag{3}$$

Assume that the queries and keys are unit vectors, then the attention computation in Eq. (2) can be approximated by

$$\text{Attention}(\mathbf{q}_t, \{\mathbf{k}_i\}, \{\mathbf{v}_i\}) \approx \sum_i \frac{\phi\left(\mathbf{q}_t\right)^\top \phi\left(\mathbf{k}_i\right) \mathbf{v}_i}{\sum_j \phi\left(\mathbf{q}_t\right)^\top \phi\left(\mathbf{k}_j\right)} \tag{4a}$$

$$= \frac{\phi\left(\mathbf{q}_t\right)^\top \sum_i \phi\left(\mathbf{k}_i\right) \otimes \mathbf{v}_i}{\phi\left(\mathbf{q}_t\right)^\top \sum_j \phi\left(\mathbf{k}_j\right)}, \tag{4b}$$

where $\otimes$ refers to the outer product. Recent works have shown that kernel-based linear attentions perform favorably against the original softmax attention on machine translation [45] and protein sequence modeling [9]. However, although the complexity is reduced into linear, the intensive floating-point multiplications in Eq. (4b) still consume a large amount of energy, which can quickly drain the batteries on mobile/edge platforms.

### 3.3 Binary Quantization

Following [49], binary quantization typically estimates the full-precision $\mathbf{u} \in \mathbb{R}^n$ using a binary $\hat{\mathbf{u}} \in \{+1, -1\}^n$ and a scaling factor $\alpha \in \mathbb{R}^+$ such that $\mathbf{u} \approx \alpha\hat{\mathbf{u}}$ holds. To find an accurate estimation, existing methods [49, 42, 34, 2, 46] minimize the quantization error as

$$\alpha^*, \hat{\mathbf{u}}^* = \arg\min \|\mathbf{u} - \alpha\hat{\mathbf{u}}\|. \tag{5}$$

By solving Problem (5), we have $\hat{\mathbf{u}} = \text{sign}(\mathbf{u})$ and $\alpha = \frac{1}{n}\|\mathbf{u}\|_{\ell 1}$, where $\text{sign}(u)$ returns 1 if $u \geq 0$ and -1 if $u < 0$. Since the sign function is non-differentiable, the straight-through estimator (STE) [5] is applied to approximate the gradient such as using the gradient of hard $\tanh$ [25] or piecewise polynomial function [39].

# 4 Proposed Method

To reduce the energy consumption of self-attention, one may perform binary quantization [46, 2] on the queries $\{\mathbf{q}_t\}_{t=1}^{N}$ and keys $\{\mathbf{k}_t\}_{t=1}^{N}$. In this case, we can replace most of the energy-expensive multiplications with the energy-efficient bit-wise operations. However, existing binary quantization methods only focus on minimizing the quantization error between the original full-precision values and the binary ones as in Eq. (5), which fails to preserve the pairwise semantic similarity between different tokens in attention, leading to performance degradation.

Note that the attention can be seen as applying kernel smoother over pairwise tokens where the kernel scores denote the similarity of the token pairs, as mentioned in Section 3.2. Motivated by this, we propose a new binarization method that applies kernelized hashing with Gaussian RBF to map the original high-dimensional queries/keys to low-dimensional similarity-preserving binary codes in Hamming space. The proposed framework, which we dub *EcoFormer*, is depicted in Figure 1 (c). To maintain the semantic similarity in attention, we learn the hash functions in a self-supervised manner. By exploiting the associative property of the linear dot-product between binary codes and the equivalence between the code inner products (*i.e.*, Hamming affinity) and the Hamming distances, we are able to approximate the self-attention in linear time with low energy cost. In the following, we first introduce the kernelized hashing attention in Section 4.1 and then show how to learn the hash functions in a self-supervised way in Section 4.2.

## 4.1 Kernelized Hashing Attention

Before applying hash functions, we let the queries $\{\mathbf{q}_t\}_{t=1}^{N}$ and keys $\{\mathbf{k}_t\}_{t=1}^{N}$ to be identical following [29, 41]. In this way, we can then apply kernelized hash functions $H : \mathbb{R}^{D_p} \mapsto \{1, -1\}^b$ without explicitly applying transformation $\phi(\cdot)$ mentioned in Section 3.2 to map $\mathbf{q}_i$ and $\mathbf{k}_j$ into $b$-bit binary codes $H(\mathbf{q}_i)$ and $H(\mathbf{k}_j)$, respectively (see Section 4.2). In this case, the Hamming distance between them can be defined as

$$\mathcal{D}\left(H(\mathbf{q}_i), H(\mathbf{k}_j)\right) = \sum_{r=1}^{b} \mathbb{1}\{H_r(\mathbf{q}_i) \neq H_r(\mathbf{k}_j)\}, \tag{6}$$

where $H_r(\cdot)$ is the $r$-th bit of the binary codes; $\mathbb{1}\{A\}$ is an indicator function that returns 1 if $A$ is satisfied and otherwise returns 0. With $\mathcal{D}\left(H(\mathbf{q}_i), H(\mathbf{k}_j)\right)$, the codes inner product between $H(\mathbf{q}_i)$ and $H(\mathbf{k}_j)$ can be formulated as

$$H(\mathbf{q}_i)^{\top} H(\mathbf{k}_j) = \sum_{r=1}^{b} \mathbb{1}\{H_r(\mathbf{q}_i) = H_r(\mathbf{k}_j)\} - \sum_{r=1}^{b} \mathbb{1}\{H_r(\mathbf{q}_i) \neq H_r(\mathbf{k}_j)\}$$

$$= b - 2\sum_{r=1}^{b} \mathbb{1}\{H_r(\mathbf{q}_i) \neq H_r(\mathbf{k}_j)\} = b - 2\mathcal{D}\left(H(\mathbf{q}_i), H(\mathbf{k}_j)\right). \tag{7}$$

Importantly, Eq. (7) shows the equivalence between the Hamming distance and the codes inner product since there is a one-to-one correspondence. By substituting with the hashed queries and keys in Eq. (4a), we can approximate the self-attention as

$$\text{Attention}(\mathbf{q}_t, \{\mathbf{k}_i\}, \{\mathbf{v}_i\}) \approx \sum_i \frac{H(\mathbf{q}_t)^{\top} H(\mathbf{k}_i)\mathbf{v}_i}{\sum_j H(\mathbf{q}_t)^{\top} H(\mathbf{k}_j)}. \tag{8}$$

Note that $H(\mathbf{q}_t)^{\top} H(\mathbf{k}_j) \in [-b, b]$. To avoid zero in denominator, we introduce a bias term $2^c$ to each inner product so that $H(\mathbf{q}_t)^{\top} H(\mathbf{k}_j) + 2^c > 0$, having no effect on the similarity measure. Here, we can simply set $c$ to $\lceil \log_2(b+1) \rceil$ where $\lceil u \rceil$ returns the least integer greater than or equal to $u$. Using the associative property of matrix multiplication, we approximate the self-attention as

$$\text{Attention}(\mathbf{q}_t, \{\mathbf{k}_i\}, \{\mathbf{v}_i\}) \approx \sum_i \frac{\left(H(\mathbf{q}_t)^{\top} H(\mathbf{k}_i) + 2^c\right)\mathbf{v}_i}{\sum_j \left(H(\mathbf{q}_t)^{\top} H(\mathbf{k}_j) + 2^c\right)}$$

$$= \frac{H(\mathbf{q}_t)^{\top} \sum_i H(\mathbf{k}_i) \otimes \mathbf{v}_i + \sum_i 2^c \mathbf{v}_i^{\top}}{H(\mathbf{q}_t)^{\top} \sum_j H(\mathbf{k}_j) + 2^c N}. \tag{9}$$

In practice, the multiplications between the binary codes and the full-precision values in Eq. (9) can be replaced by simple additions and subtractions, which greatly reduce the computational overhead in terms of on-chip energy footprint. Moreover, the multiplications with a powers-of-two $2^c$ can also be implemented by efficient *bit-shift* operations. As a result, the only multiplications come from the element-wise divisions between the numerator and denominator.

## 4.2 Self-supervised Hash Function Learning

Given queries $\mathcal{Q} = \{\mathbf{q}_1, \ldots, \mathbf{q}_N\} \subset \mathbb{R}^{D_p}$, we seek to learn a group of hash functions $h : \mathbb{R}^{D_p} \mapsto \{1, -1\}$. Instead of explicitly applying the transformation function $\phi(\cdot)$ mentioned in Section 3.2, we compute the hash functions with a kernel function $\kappa(\mathbf{q}_i, \mathbf{q}_j) : \mathbb{R}^{D_p} \times \mathbb{R}^{D_p} \mapsto \mathbb{R}$. Given $\mathbf{Q} = [\mathbf{q}_1, \cdots, \mathbf{q}_N]^\top \in \mathbb{R}^{N \times D_p}$, we randomly sample $m$ queries $\mathbf{q}_{(1)}, \ldots, \mathbf{q}_{(m)}$ from $\mathcal{Q}$ as support samples following the kernel-based supervised hashing (KSH) [36] and define a hash function $h$ as

$$h(\mathbf{Q}) = \text{sign}\left(\sum_{j=1}^m \left(\kappa\left(\mathbf{q}_{(j)}, \mathbf{Q}\right) - \mu_j\right) a_j\right) = \text{sign}\left(\mathbf{g}(\mathbf{Q})\mathbf{a}\right), \tag{10}$$

where $\mathbf{a} = [a_1, \cdots, a_m]^\top$ is the weight of $h$, $\mu_j = \frac{1}{n}\sum_{i=1}^N \kappa\left(\mathbf{q}_{(j)}, \mathbf{q}_i\right)$ is to normalize the kernel function to zero-mean, and $\mathbf{g} : \mathbb{R}^{D_p} \mapsto \mathbb{R}^m$ is a mapping defined by $\mathbf{g}(\mathbf{Q}) = \left[\kappa\left(\mathbf{q}_{(1)}, \mathbf{Q}\right) - \mu_1, \ldots, \kappa\left(\mathbf{q}_{(m)}, \mathbf{Q}\right) - \mu_m\right] \in \mathbb{R}^{N \times m}$. Then, we define the kernelized hash function $H(\cdot)$ as

$$H(\mathbf{Q}) = [h_1(\mathbf{Q}), \cdots, h_b(\mathbf{Q})] = \begin{bmatrix} h_1(\mathbf{q}_1), \cdots, h_b(\mathbf{q}_1) \\ \cdots\cdots \\ h_1(\mathbf{q}_N), \cdots, h_b(\mathbf{q}_N) \end{bmatrix} = \text{sign}\left(\mathbf{g}(\mathbf{Q})\mathbf{A}\right), \tag{11}$$

where $\mathbf{A} = [\mathbf{a}_1, \cdots, \mathbf{a}_b] \in \mathbb{R}^{m \times b}$, and $h_r(\mathbf{Q}) = \text{sign}\left(\mathbf{g}(\mathbf{Q})\mathbf{a}_r\right)$ is the hash function for the $r$-th bit.

To guide the learning of the binary codes, we hope that similar token pairs will have the minimal Hamming distance while dissimilar token pairs will have the maximal distance. Nevertheless, directly optimizing the Hamming distance is difficult due to the non-convex and non-smooth formulation in Eq. (6). Utilizing the equivalence between the code inner products and the Hamming distances in Eq. (7), we instead conduct optimization based on the Hamming affinity to minimize the reconstruction error as

$$\min_{\mathbf{A}} \left\| H(\mathbf{Q})H(\mathbf{Q})^\top - b\mathbf{Y} \right\|_F^2 = \min_{\mathbf{A}} \left\| \sum_{r=1}^b h_r(\mathbf{Q})h_r(\mathbf{Q})^\top - b\mathbf{Y} \right\|_F^2, \tag{12}$$

where $\|\cdot\|_F$ is the Frobenius norm and $\mathbf{Y} \in \mathbb{R}^{N \times N}$ is the target Hamming affinity matrix. To preserve the similarity relations between queries and keys, we use the attention scores as the *self-supervised* information to construct $\mathbf{Y}$. Let $\mathcal{S}$ and $\mathcal{U}$ be the similar and dissimilar pairs of tokens. We obtain $\mathcal{S}$ and $\mathcal{U}$ by selecting the token pairs with the Top-$l$ largest and smallest attention scores. We then construct pairwise labels $\mathbf{Y}$ as

$$\mathbf{Y}_{ij} = \begin{cases} 1, & (\mathbf{q}_i, \mathbf{q}_j) \in \mathcal{S} \\ -1, & (\mathbf{q}_i, \mathbf{q}_j) \in \mathcal{U} \\ 0, & \text{otherwise.} \end{cases} \tag{13}$$

However, Problem (12) is NP-hard. To solve it efficiently, we adapt discrete cyclic coordinate descent to learn binary codes sequentially. Specifically, we only solve $\mathbf{a}_r$ once the previous $\mathbf{a}_1, \cdots, \mathbf{a}_{r-1}$ have been optimized. Let $\hat{\mathbf{Y}}_{r-1} = b\mathbf{Y} - \sum_{t=1}^{r-1} h_t(\mathbf{Q})h_t(\mathbf{Q})^\top$ be the residual matrix, where $\hat{\mathbf{Y}}_0 = b\mathbf{Y}$. Then, we can minimize the following objective to obtain $\mathbf{a}_r$

$$\min_{\mathbf{a}_r} \left\| h_r(\mathbf{Q})h_r(\mathbf{Q})^\top - \hat{\mathbf{Y}}_{r-1} \right\|_F^2 = \min_{\mathbf{a}_r} -2h_r(\mathbf{Q})^\top \hat{\mathbf{Y}}_{r-1} h_r(\mathbf{Q}) + C, \tag{14}$$

where $C = \left(h_r(\mathbf{Q})^\top h_r(\mathbf{Q})\right)^2 + \text{tr}\left(\hat{\mathbf{Y}}_{r-1}\right)$ is a constant. Note that $\hat{\mathbf{Y}}_{r-1}$ is a symmetric matrix.

Therefore, Problem (14) is a standard binary quadratic programming problem, which can be efficiently solved by many existing methods, such as the LBFGS-B solver [69] and block graph cuts [33]. To learn $\mathbf{a}_r$ in conjunction with network parameters, we propose to solve Problem (14) using the gradient-based methods. For the non-differentiable sign function, we use STE [5] to approximate the gradient using hard $\texttt{tanh}$ as mentioned in Section 3.3. Note that learning the hash functions for each epoch is computationally expensive yet unnecessary. We only learn the hash functions per $\tau$ epoch.

Table 2: Main results on ImageNet-1K. The number of multiplications, additions, as well as on-chip energy consumption are calculated based on an image of resolution $224 \times 224$. The throughput is measured with a mini-batch size of 32 on a single NVIDIA RTX 3090 GPU.

| Model | Method | #Mul. (B) | #Add. (B) | Energy (B pJ) | Throughput (images/s) | Top-1 Acc. (%) |
|---|---|---|---|---|---|---|
| PVTv2-B0 [62] | MSA | 2.02 | 1.99 | 9.25 | 850 | 70.77 |
| | **Ours** | **0.54** | **0.56** | **2.49** | **1379** | **70.44** |
| PVTv2-B1 | MSA | 5.02 | 5.00 | 23.07 | 621 | 78.83 |
| | **Ours** | **2.03** | **2.09** | **9.39** | **874** | **78.38** |
| PVTv2-B2 | MSA | 8.64 | 8.60 | 39.71 | 404 | 81.82 |
| | **Ours** | **3.85** | **3.97** | **17.82** | **483** | **81.28** |
| PVTv2-B3 | MSA | 11.86 | 11.82 | 54.56 | 310 | 82.26 |
| | **Ours** | **6.54** | **6.72** | **30.25** | **325** | **81.96** |
| PVTv2-B4 | MSA | 15.97 | 15.93 | 73.43 | 247 | 82.42 |
| | **Ours** | **9.57** | **9.82** | **44.25** | **249** | **81.90** |
| Twins-SVT-S [10] | MSA | 5.96 | 5.91 | 27.36 | 426 | 81.66 |
| | **Ours** | **2.72** | **2.81** | **12.59** | **576** | **80.22** |

# 5 Experiments

## 5.1 Comparisons on ImageNet-1K

To investigate the effectiveness of the proposed method, we conduct experiments on ImageNet-1K [30], a large-scale image classification dataset that contains ~1.2M training images from 1K categories and 50K validation images. We compare our kernelized hashing attention with standard MSA by adapting the two attention approaches into two popular vision Transformer frameworks PVTv2 [62] and Twins [10]. We measure model performance by the Top-1 accuracy. Furthermore, as FLOPs cannot accurately reflect the computational cost in our proposed method, we measure the model complexity by the number of multiplications and additions, separately, as done in [53]. Specifically, we calculate FLOPs following [61], where we count the multiply-accumulate operations for all layers. In this case, each multiply-accumulate operation consists of an addition and a multiplication. We also count the multiplications in the scaling operations. Moreover, we report the on-chip energy consumption according to Table 1 and the throughput with a mini-batch size of 32 on a single NVIDIA RTX 3090 GPU.

**Implementation details.** All training images are resized to $256 \times 256$, and $224 \times 224$ patches are randomly cropped from an image or its horizontal flip, with the per-pixel mean subtracted. To obtain the MSA baselines, we first replace the original attention layers in PVTv2 [62] and Twins [10] with standard MSAs and initialize the models with the pretrained weights on ImageNet-1K. Next, we finetune each model on ImageNet-1K with 100 epochs. Based the pretrained MSA weights, we then apply our approach to each model and finetune on ImageNet-1K with 30 epochs. All models in this experiment are trained on 8 V100 GPUs with a total batch size of 256. We set the initial learning rate to $2.5 \times 10^{-5}$ for PVTv2 and $1.25 \times 10^{-4}$ for Twins. We use AdamW optimizer [40] with a cosine decay learning rate scheduler. All other hyperparameters are the same as in PVTv2. Also note that recent hierarchical ViTs [62, 38, 10] have multiple stages to incorporate pyramid feature maps. At the last stage, they usually apply standard MSAs due to the comparably low-resolution feature maps. This design is also adopted in PVTv2 and Twins. Therefore, we follow the common practice and do not modify the attention layers at the last stage. For the hash functions learning, we set the number of support samples $m$ and update interval $\tau$ to 25 and 30, respectively. The hyper-parameter $l$ in constructing pairwise labels $\mathbf{Y}$ is set to 10. We set the hash bit $b$ to 16.

**Results analysis.** We report the results in Table 2. In general, our baseline MSA has more multiplications than additions. In contrast, our EcoFormer replaces most of the floating-point multiplications in attention with simple additions. Therefore, there are more additions than multiplications in our EcoFormer. Compared to MSA, our method achieves lower computational complexity, less energy consumption and higher throughput with comparable performance. For example, based on PVTv2-B0, our method saves around 73% multiplications and 72% additions, as well as reducing 73% on-chip energy consumption, which demonstrates the energy-efficiency of our approach. With more efficient accumulation implementation, the throughput of our EcoFormer can be further improved. Besides, a larger model comes with a larger proportion of computational and on-chip energy cost dominated by FFNs, as shown in Figure 2. In this case, as our approach focuses on the attention layers, the energy-

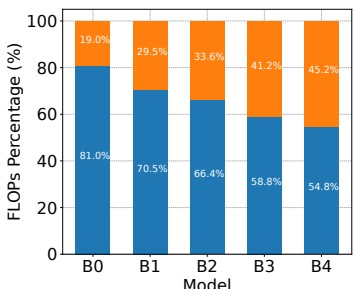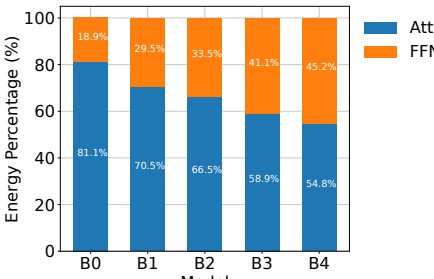

Figure 2: FLOPs and on-chip energy footprint percentage of attention layers (Attn) and feed-forward layers (FFN) in different variants of PVTv2 with standard MSAs. For a bigger model, FFN takes a larger proportion of the computational cost and on-chip energy footprint.

saving from larger models is comparably less than smaller models (*e.g.*, PVTv2-B0 vs. PVTv2-B4). Nonetheless, we still reduce significant computational cost and on-chip energy consumption. For example, on PVTv2-B4, we save around 40% on-chip energy consumption. Compared with PVTv2, the performance drop of our method is slightly larger on Twins. One possible reason is that our method may be sensitive to the conditional positional encodings in Twins.

## 5.2 Comparisons on Long Range Arena

To evaluate the performance of different efficient attentions under long-context scenarios, we train our EcoFormer on two tasks, **Text** and **Retrieval** from the Long Range Arena (LRA) benchmark [55] following the settings of [70]. Our implementations are based on the released code of [64]. We use the same hyper-parameters $m$, $\tau$ and $b$ as in ImageNet-1K experiments. We show the results in Table 3. From the table, our EcoFormer achieves comparable performance with much lower on-chip energy consumption. For example, on **Text**, compared with MSA, our method saves around 94.6% multiplications and 93.7% additions as well as 94.5% on-chip energy consumption, which is more efficient than existing attention mechanisms.

Table 3: Comparisons of different methods on Long Range Arena (LRA). We report the classification accuracy (%) for **Text** as well as **Retrieval** and average accuracy across two tasks. * denotes that we obtain the results from the original paper.

| Method | #Mul. (B) | #Add. (B) | Energy (B pJ) | Text (4K) | Retrieval (4K) | Average |
|---|---|---|---|---|---|---|
| Transformer | 4.63 | 4.57 | 21.25 | 64.87 | 79.62 | 72.25 |
| Performer [9] | 0.83 | 0.84 | 3.83 | 64.82 | 79.08 | 71.95 |
| Linformer [60] | 0.81 | 0.81 | 3.74 | 57.03 | 78.11 | 67.57 |
| Reformer [29] | 0.54 | 0.54 | 2.49 | 65.19 | 79.46 | 72.33 |
| Combiner* [50] | 0.51 | 0.51 | 2.34 | 64.36 | 56.10 | 60.23 |
| **EcoFormer** | **0.25** | **0.29** | **1.17** | 64.79 | 78.67 | 71.73 |

## 5.3 Ablation Study

In this section, we evaluate the effectiveness of our EcoFormer by comparing it with different binarization approaches and efficient attention mechanisms. By default, we train each model from scratch on CIFAR-100 with 2 GPUs for 300 epochs. The total batch size is 64. The initial learning rate is $6.25 \times 10^{-6}$. For the hash functions learning, we set update interval $\tau$ to 300. All the other hyperparameters are the same as in ImageNet-1K experiments.

Table 4: Performance comparisons with different binarization methods on CIFAR-100.

| Model | Method | #Mul. (B) | #Add. (B) | Energy (B pJ) | Top-1 Acc. (%) |
|---|---|---|---|---|---|
| | FP-EcoFormer | 0.94 | 0.94 | 4.33 | 70.78 |
| PVTv2-B0 | Bi-EcoFormer | 0.63 | 0.83 | 3.09 | 70.06 |
| | **EcoFormer** | **0.54** | **0.56** | **2.49** | **71.23** |
| | FP-EcoFormer | 5.96 | 5.91 | 27.36 | 80.04 |
| Twins-SVT-S | Bi-EcoFormer | 3.01 | 3.59 | 14.38 | 80.04 |
| | **EcoFormer** | **2.72** | **2.81** | **12.58** | **80.31** |

**Quantization vs. hashing.** To investigate the effect of different binarization methods, we compare our EcoFormer with the following methods: **FP-EcoFormer**: Based on EcoFormer, we do not

binarize queries and keys in attentions. **Bi-EcoFormer**: Relying on EcoFormer, we use the same binary quantization [25] as BinaryBERT [2] and BiBERT [46] to obtain binarized queries and keys instead of our proposed hash functions. For fair comparisons, the attention operations in the compared method are in linear complexity. We apply different methods to PVTv2-B0 and Twins-SVT-S and report the results in Table 4. We observe that our EcoFormer consistently outperforms Bi-EcoFormer on different frameworks. For example, based on PVTv2-B0, our EcoFormer surpasses Bi-EcoFormer by 1.17% in terms of the Top-1 accuracy. Compared with binary quantization, our proposed self-supervised hash functions preserve the pairwise similarity of attention, leading to better performance. Moreover, our EcoFormer does not need to explicitly compute transformation $\phi(\cdot)$ as in Eq. (4b). Therefore, the energy cost of our EcoFormer is lower than Bi-EcoFormer.

Table 5: Performance comparisons with different hash functions regarding PVTv2-B0 on CIFAR-100.

| Method | #Mul. (B) | #Add. (B) | Energy (B pJ) | Top-1 Acc. (%) |
|---|---|---|---|---|
| LSH-EcoFormer | 0.68 | 0.69 | 3.12 | 70.18 |
| KLSH-EcoFormer | 0.54 | 0.56 | 2.49 | 70.66 |
| **EcoFormer** | **0.54** | **0.56** | **2.49** | **71.23** |

**Effect of different hash functions.** To investigate the effect of different hash functions, we include the following methods for comparisons: **LSH-EcoFormer**: Relying on EcoFormer, we use Locality-Sensitive Hashing (LSH) [13] rather than our proposed kernelized hash function. **KLSH-EcoFormer**: Based on EcoFormer, we replace the proposed hash function with Kernelized Locality-Sensitive Hashing (KLSH) [31]. We report the results in Table 5. We can observe that KLSH-EcoFormer outperforms LSH-EcoFormer by 0.48% in terms of the Top-1 accuracy with less on-chip energy consumption. The reason can be attributed to that LSH is based on random linear projection, which can not deal with the non-linear softmax attention well. Critically, our EcoFormer further improves the performance by 0.57% on the Top-1 accuracy. Compared with random hashing in KLSH, our EcoFormer learns the hash functions with additional self-supervised information. Therefore, our learned binary codes are better at preserving the token similarity.

Table 6: Comparison with other efficient attention methods regarding PVTv2-B0 [62] on CIFAR-100.

| Method | #Mul. (B) | #Add. (B) | Energy (B pJ) | Top-1 Acc. (%) |
|---|---|---|---|---|
| Transformer | 2.02 | 1.99 | 9.25 | 71.44 |
| Performer [9] | 0.94 | 0.94 | 4.33 | 70.78 |
| Linformer [60] | 0.69 | 0.69 | 3.18 | 71.17 |
| Reformer [29] | 1.62 | 1.63 | 7.44 | 70.56 |
| **EcoFormer** | **0.54** | **0.56** | **2.49** | **71.23** |

**Comparing with other efficient attention mechanisms.** To compare EcoFormer with different attention mechanisms, we conduct experiments on CIFAR-100 based on PVTv2-B0 in Table 6. In our experiments, we directly replace the attention layers with each compared method in PVTv2-B0 [62]. In general, compared to other efficient attention mechanisms, EcoFormer saves more computations and reduces more on-chip energy consumption while achieving better performance. Particularly, benefiting from the multiplication-saving operations and low-dimensional binary queries and keys, EcoFormer saves more on-chip energy than Performer.

Table 7: Latency and energy comparisons with different attention methods. We measure the latency and energy of an attention layer with a batch size of 16, a sequence length of 3,136 and an embedding dimension of 32 on a BitFusion [51] simulator.

| Method | Latency (ms) | Energy (pJ) |
|---|---|---|
| Transformer | 0.0036 | 85,692.18 |
| Performer [9] | 0.0019 | 41,113.64 |
| Linformer [60] | 0.0018 | 45,770.61 |
| Reformer [29] | 0.0024 | 57,305.47 |
| **EcoFormer** | **0.0010** | **24,990.75** |

Also note that since the proposed kernelized hash function $H(\cdot)$ does not need to explicitly apply transformation $\phi(\cdot)$ to the queries and keys as in Eq. (4b), EcoFormer simultaneously reduces more multiplications and additions than Performer. Besides, Linformer achieves competitive results. However, as the size of the learnable low-rank projection parameters depends on the length of the input sequence, Linformer is not scalable to different image resolutions, whereas EcoFormer with sufficient bits is agnostic to the sequence length.

**Latency and energy on BitFusion [51].** To show the actual energy consumption and latency, we test different methods on a simulator of BitFusion, a bit-flexible microarchitecture synthesized in 45 nm technology. From Table 7, EcoFormer shows much lower latency and on-chip energy than the other efficient attention methods, which further verifies the advantage of EcoFormer.

**Effect of training from scratch on ImageNet-1K.** To explore the effect of training from scratch, we apply EcoFormer to PVTv2-B0 and PVTv2-B1. We follow the experimental settings mentioned in Section 5.1 except that we train the model from scratch with 300 epochs. The initial learning rate is set to $2.5 \times 10^{-4}$. From Table 8, EcoFormer achieves comparable performance while significantly reducing the computational complexity and on-chip energy consumption. The accuracy drop from discretization comes from the gradient approximation for the non-differentiable sign function, which can be mitigated by more advanced optimization methods, such as regularization [16], knowledge distillation [43, 46], relaxed optimization [23, 3], appending full-precision branches [39, 42], *etc.*

Table 8: Performance comparisons of different methods on ImageNet-1K. All the models are trained from scratch. The number of multiplications, additions, and on-chip energy consumption are calculated based on an image of resolution $224 \times 224$.

| Model | Method | #Mul. (B) | #Add. (B) | Energy (B pJ) | Top-1 Acc. (%) |
|---|---|---|---|---|---|
| PVTv2-B0 | MSA | 2.02 | 1.99 | 9.25 | 69.72 |
| | **Ours** | **0.54** | **0.56** | **2.49** | **68.70** |
| PVTv2-B1 | MSA | 5.02 | 5.00 | 23.07 | 78.34 |
| | **Ours** | **2.03** | **2.09** | **9.39** | **77.49** |

**Effect of different $m$.** To investigate the effect of different numbers of support samples $m$, we train EcoFormer with different $m$ based on PVTv2-B0. We report the results on CIFAR-100 in Table 9. As we increase $m$, the performance becomes better along with the increase in on-chip energy consumption. For example, the model obtained with $m = 15$ outperforms that of $m = 10$ by 0.19% on the Top-1 accuracy with little additional energy cost. We speculate that, with more support samples, we can capture more accurate statistics in Eq. (10) and hence lead to better performance. Since our EcoFormer achieves the best performance with $m = 25$, we use it by default in our experiments.

Table 9: Performance comparisons with different #support samples $m$. We report the results of PVTv2-B0 on CIFAR-100.

| $m$ | #Mul. (B) | #Add. (B) | Energy (B pJ) | Top-1 Acc. (%) |
|---|---|---|---|---|
| 10 | 0.53 | 0.55 | 2.46 | 70.73 |
| 15 | 0.53 | 0.56 | 2.47 | 70.92 |
| 20 | 0.53 | 0.56 | 2.48 | 70.81 |
| 25 | 0.54 | 0.56 | 2.49 | **71.23** |

## 6 Conclusion and Future Work

In this paper, we have presented a novel energy-saving attention mechanism with linear complexity to save the vast majority of multiplications from a new binarization perspective, making the deployment of Transformer models at scale feasible on edge devices. We are inspired by the fact that conventional binarization methods are built upon statistical quantization error minimization without considering to preserve the pairwise similarity relations between tokens. To this end, we customize binarization to softmax attention by mapping the original token features into compact binary codes in Hamming space using a set of kernel-based hash functions, where the similarity can be measured by codes dot product. The hash functions for queries/keys are learned to encourage the Hamming affinity of a token pair to be close to the target obtained from the attention scores, in a self-supervised way. Extensive experiments have demonstrated that EcoFormer saves significant on-chip energy footprint while achieving comparable performance with standard attentions on ImageNet-1K, Long Range Arena and CIFAR-100. In terms of the future work, we can further binarize the value vectors in attention, multi-layer perceptrons and non-linearities in Transformer to make it fully binarized for more significant on-chip energy-saving. We may also extend EcoFormer to other NLP tasks such as machine translation and speech analysis tasks to make it more impactful to wider communities.

**Limitations and societal impact.** We have shown that EcoFormer is more energy-efficient compared to the standard attention. However, in practice, the addition operations between binary codes and floating-point numbers will require specialized GPU kernels (*e.g.*, customized CUDA operators) for further acceleration. Moreover, for the short sequence scenario, our EcoFormer suffers from severe performance drop due to the limited representational capability. Our work potentially brings some negative societal impacts that training large Transformer models requires extensive computations, resulting in financial and environmental costs. A promising solution is to jointly optimize training and inference efficiency.

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
