# Appendix

We organize our supplementary material as follows.

- In Section A, we show the complexity and energy footprint saving for the attention layers only.

- In Section B, we provide more throughput results on ImageNet-1K.

## A  Cost Saving for Attentions Only

In this section, we show the results of cost saving for attentions only using different architectures on ImageNet-1K. From Table A, our EcoFormer consistently saves massive on-chip energy footprint. In particular, on PVTv2-B0, we save 93% on-chip energy footprint. On larger models, the saving is still significant. Note that larger models also come with more computational cost from the linear projection layers in MSAs, as shown in Figure 2 in the main paper. Since our EcoFormer does not target these projection layers, larger models save slightly less energy in MSAs compared to those of smaller models.

Table A: The cost saving for attentions only, excluding other types of layers. The number of multiplications, additions, as well as on-chip energy consumption are calculated based on an image of resolution $224 \times 224$.

| Model | Method | #Mul. (B) | #Add. (B) | Energy (B pJ) |
|---|---|---|---|---|
| PVTv2-B0 [3] | MSA | 1.58 | 1.56 | 7.26 |
| | **Ours** | **0.10** | **0.13** | **0.50** (**-93%**) |
| PVTv2-B1 | MSA | 3.38 | 3.36 | 15.52 |
| | **Ours** | **0.39** | **0.45** | **1.84** (**-88%**) |
| PVTv2-B2 | MSA | 5.59 | 5.56 | 25.69 |
| | **Ours** | **0.80** | **0.92** | **3.80** (**-85%**) |
| PVTv2-B3 | MSA | 6.85 | 6.81 | 31.49 |
| | **Ours** | **1.53** | **1.71** | **7.21** (**-77%**) |
| PVTv2-B4 | MSA | 8.63 | 8.59 | 39.69 |
| | **Ours** | **2.24** | **2.49** | **10.52** (**-74%**) |
| Twins-SVT-S [2] | MSA | 4.01 | 3.96 | 18.41 |
| | **Ours** | **0.77** | **0.86** | **3.63** (**-80%**) |

## B  More Throughput Results on ImageNet-1K

To show the actual inference speed on a hardware device, we measure the throughput of different methods on a single NVIDIA RTX 3090 GPU. We compare EcoFormer with the standard multi-head self-attention (MSA) and kernel-based linear attention (KLA) [1]. From Table B, KLA shows higher throughput than MSA, while our EcoFormer achieves even faster throughput than KLA, thanks to the reduced feature dimensions ($b$ vs. $D_p$) of queries and keys. With efficient energy-efficient accumulation implementation, the throughput of our EcoFormer can be further improved, which will be explored in the future.

Table B: Throughput (images/s) of different methods on ImageNet-1K. MSA denotes the standard multi-head self-attention and KLA represents the kernel-based linear attention. The throughput is measured with a mini-batch size of 32 and an image resolution of $224 \times 224$ on a single NVIDIA RTX 3090 GPU.

| Method | PVTv2-B0 | PVTv2-B1 | PVTv2-B2 | Twins-SVT-S |
|---|---|---|---|---|
| MSA | 850 | 621 | 404 | 426 |
| KLA [1] | 1166 | 769 | 444 | 489 |
| Ours | **1379** | **874** | **483** | **576** |