# OpenReview forum: "EcoFormer: Energy-Saving Attention with Linear Complexity"
_NeurIPS.cc/2022/Conference — NeurIPS 2022 Accept_

### Official Review · Reviewer_BP1F · 2022-07-10

**Rating:** 4
**Confidence:** 4
**Soundness:** 2 fair
**Presentation:** 3 good
**Contribution:** 3 good

**Summary:**

This paper proposes a new binarization framework to customize high-dimensional softmax attention and reduce computation complexity in Transformers. Specifically, it utilizes the kernelized hashing to map the original queries and keys into low-dimensional binary codes in Hamming space. The kernelized hashing function is learned to maximize the similarities between the binarized codes and ground-truth relations in the original attention maps. Experiments on classification tasks show it achieves comparable performance with much fewer multiplications and additions.

**Questions:**

1. In the line 202, $c$ is set to $\lceil \log_2 b \rceil$, then the inequation in line 201 could be 0, right?
2. The notations in Sec. 4.2 are confusing and difficult to understand, for example, in Eq. (11), the LHS $H(q)$ should be a bx1 vector as you defined at the beginning of this subsection, but RHS is Nxb after the matrix multiplication. How come this happens?


**Limitations:**

The additional parameters of weights $a$ are optimized iteratively in the training process, the computation cost and time cost might still be a problem.

**Strengths And Weaknesses:**

Strengths:
1. The motivation of this work is promising and binarizing the Transformers through kernelized hashing functions in a self-supervised way is novel.
2. In this paper, authors explicit the idea and methods clearly and easy to follow how they solve the whole optimization problem.
3. Experimental results on ImageNet show high efficiency compared with the baselines.

Weaknesses:
1. Sec. 4.2 seems to have some inconsistent size of matrices or vectors, which makes this key part hard to follow and misunderstanding.
2. There are some related works using binarization strategy in Transformers, too, including the latest BiT [1]. etc. However, there is no comparison with these methods.
3. In the section of experiments, tables show the number of additions and multiplications but the details of how to calculate them are missed.

[1] Liu, Zechun, et al. "BiT: Robustly Binarized Multi-distilled Transformer." arXiv preprint arXiv:2205.13016 (2022).

---

> ### Author Response · Authors · 2022-08-02
> **Response to Reviewer BP1F**
>
> Thanks for your valuable comments.
>
> **Q1.** No comparisons with binary transformers, such as [A].
>
> **A1.** As mentioned in L129-133 and Section 3.3, we have discussed many binary transformers, including BinaryBERT [2] and BiBERT [47], which may not well preserve the similarity relations among tokens. We do not compare with [A] since [A] is **an ArXiv paper online on 05/25/2022, which is well after the submission deadline**. Compared with these methods, our EcoFormer customizes a new binarization paradigm to softmax attention from the kernelized hashing perspective, which helps to preserve the pairwise similarity while significantly saving energy cost.
>
> To demonstrate this, as mentioned in L296-307 (L288-299 in the previous version), we have compared our EcoFormer with Quant-EcoFormer which uses the same binary quantization method as BinaryBERT and BiBERT except that we do not use any advanced training scheme such as knowledge distillation. We have shown the results in Table 4 (Table 3 in the previous version). From the table, our EcoFormer consistently outperforms Quant-EcoFormer on different frameworks while saving more energy. To make it clear, we have changed "Quant-EcoFormer" to "Bi-EcoFormer" in our revised manuscript.
>
> **Q2.** How to calculate the number of additions and multiplications?
>
> **A2.** As mentioned in L249-250, we calculate the number of additions and multiplications to measure the model complexity following [54]. Specifically, we calculate floating-point operations (FLOPs) following [62], where we count the multiply-accumulate operations for all layers. In this case, each multiply-accumulate operation consists of an addition and a multiplication. We also count the multiplications in the scaling operations. Therefore, our baseline MSA has more multiplications than additions. For our EcoFormer, as mentioned in L70-71, we can replace most of the floating-point multiplications in attention with simple additions. Therefore, there are more additions than multiplications in our EcoFormer. We have included these descriptions in Section A of the supplementary file.
>
> **Q3.** In line 202, $c$ is set to $⌈log_2⁡b⌉$, then the inequation in line 201 could be 0, right?
>
> **A3.** Thanks for pointing it out. In our revision, we set $c$ to $⌈log_2⁡ (b+1)⌉$.
>
> **Q4.** Sec 4.2 seems to have some inconsistent sizes of matrices or vectors, which makes this key part hard to follow and misunderstand. For example, in Eq. (11), the LHS $H({\bf q})$ should be a $b \times 1$ vector as you defined at the beginning of this subsection, but RHS is $N \times b$ after the matrix multiplication.
>
> **A4.** We would like to clarify that our notations do not have inconsistent sizes. In Eq. (11), $H({\bf q})$ is a matrix with a shape of $N \times b$ since ${\bf q} = [{\bf q}_1, \cdots, {\bf q}_N]^{\top}$ is a matrix with a shape of $N \times D_p$ , where ${\bf q}_i \in \mathbb{R}^{D_p}$ is a vector as defined at the beginning of Section 4.2. To avoid misunderstanding, in our revision, we have changed ${\bf q}$ to ${\bf Q}$, ${\bf G}$ to ${\bf g}({\bf Q})$ and expanded Eq. (11) to
>
> $ H({\bf Q})= \left[ h_1({\bf Q}), \cdots, h_b({\bf Q})  \right] = \left[ \begin{array}{c}
>      h_1({\bf q}_1), \cdots, h_b({\bf q}_1) \\\\
>      \cdots~\cdots \\\\
>      h_1({\bf q}_N), \cdots, h_b({\bf q}_N)
> \end{array}
> \right] =
> \mathrm{sign}\left({\bf g}({\bf Q}){\bf A}\right).$
>
> **Q5.** The additional parameters of weights $a$ are optimized iteratively in the training process, the computation cost and time cost might still be a problem.
>
> **A5.** As mentioned in L240-241, we only learn the hash functions per $\tau$ epoch to prevent the prohibitive computational cost. Heuristically, as mentioned in L265-266, we find that setting $\tau$ to 30 achieves good performance with only a small amount of additional cost (e.g., 156 seconds for PVTv2-B4). Moreover, we target at improving the inference efficiency, rather than the training efficiency.
>
> **Reference**
>
> [A] BiT: Robustly Binarized Multi-distilled Transformer. arXiv 2022.

---

> ### Author Response · Authors · 2022-08-07
> **Request for discussion**
>
> Dear Reviewer BP1F,
>
> As the rebuttal discussion is about to end soon, please don’t hesitate to let us know if there are still some concerns/questions. We have addressed your primary concerns regarding comparing with binary Transformers and made Eq. (11) easier to understand. We sincerely thank you again for your great efforts in reviewing this paper.
>
> Best regards,
>
> Authors of #231

---

### Official Review · Reviewer_M9L7 · 2022-07-11

**Rating:** 6
**Confidence:** 5
**Soundness:** 3 good
**Presentation:** 3 good
**Contribution:** 3 good

**Summary:**

This paper proposes a learnable kernelized hashing to binarize the queries and keys such that the hamming distance between binary codes matches the dot-product similarity relations in the original attention maps. The hashing functions use self-supervised learning using the targets from the most similar and dissimilar pair of queries and keys.  The paper then exploits the equivalence between the hamming distance and dot product of binary codes to approximate the attention in linear complexity with respect to the sequence length.  More importantly, the binarized codes allow replacing the power-hungry floating-point multiplications with simple additions and subtractions. Experiments on vision tasks show that the proposed method results in 50-80% energy and chip area savings for a minor accuracy loss.


**Questions:**

**Questions and Suggestions**
  - In lines 254-257, the paper describes the setup for the ImageNet experiments. The proposed method starts from the weights of the pre-trained MSA model and fine-tunes it for 30 epochs. In contrast,  for CIFAR experiments, the models were trained from scratch. Why was the ImageNet model not trained from scratch?
  - Besides Reformer, other works have looked at LSH-based schemes to identify similar pairs of queries and keys. A discussion contrasting the current work from these could be included in the related works:
    - SMYRF: Efficient Attention using Asymmetric Clustering (NeurIPS 2020)
    - Fast Transformers with Clustered Attention (NeurIPS 2020)
    - Sparse Attention with Learning to Hash (ICLR 2022)

**Limitations:**

These are discussed.

**Strengths And Weaknesses:**

**Strengths**
  - The paper is well written. The language is clear and the flow of logic makes the paper easy to understand.
  - The idea of using learnable kernelized hashing to preserve similarity relations of binary codes is novel. The results are encouraging and demonstrate the effectiveness of the proposed methods on the vision tasks.

**Weakness**
  - The paper demonstrates the efficacy of the proposed methods on vision tasks. I am concerned about the accuracy on natural language processing and other tasks involving sparse attention patterns where the sensitivity of the binarized attention codes could be affected. Long-range arena [1] could be a good benchmark to test the accuracy for various tasks.
  - The paper currently reports the number of additions and multiplications. However, a discussion on how these numbers were derived is missing. It would be good to include this (possibly in the supplementary).
  - The paper mentions that the Adder Attention [2] results in a drastic performance drop. However, the paper doesn't offer any evidence of this in a comparable setting. From the Adder Attention paper, the DeiT-B model with additions can drop the top-1 accuracy from 81.8%  to 80.4%, similar to the accuracy drop for the proposed method with the Twins-SVT-S model. Thus, a fair comparison would be to use identical networks to compare the accuracy and performance benefits.

[1] Long Range Arena: A Benchmark for Efficient Transformers

[2] Adder Attention for Vision Transformer

---

> ### Author Response · Authors · 2022-08-02
> **Response to Reviewer M9L7 (Part 1)**
>
> Thanks for your constructive comments.
>
> **Q1.** More results on natural language processing, such as Long-range arena [A].
>
> **A1.** To evaluate the performance of different methods under long-context scenarios, we train our EcoFormer on two tasks, **Text** and **Retrieval** from the Long Range Arena (LRA) benchmark [A] following the settings of [B]. Our implementations are based on the released code of [C]. From Table III, our EcoFormer achieves comparable performance with much lower energy consumption. For example, on **Text**, compared with standard Transformer, our method saves around 94.6% multiplications and 93.7% additions as well as 94.5% energy consumption, which is more efficient than the existing attention mechanisms. These results further justify the effectiveness of our EcoFormer under long-context scenarios.
>
> **Q2.** How to calculate the number of additions and multiplications?
>
> **A2.** As mentioned in L249-250, we calculate the number of additions and multiplications to measure the model complexity following [54]. Specifically, we calculate floating-point operations (FLOPs) following [62], where we count the multiply-accumulate operations for all layers. In this case, each multiply-accumulate operation consists of an addition and a multiplication. We also count the multiplications in the scaling operations. Therefore, our baseline MSA has more multiplications than additions. For our EcoFormer, as mentioned in L70-71, we can replace most of the floating-point multiplications in attention with simple additions. Therefore, there are more additions than multiplications in our EcoFormer. We have included these descriptions in Section A of the supplementary file.
>
> **Q3.** Why not compare with Adder Attention [54] on DeiT?
>
> **A3.** First, as mentioned in L51-62, our EcoFormer focuses on improving the efficiency of attention via kernelized hashing for the long sequence scenario. Therefore, we do not apply our EcoFormer to DeiT due to the short sequence length of 196. Second, as the source code of Adder Attention is unavailable, we are unable to compare EcoFormer with Adder Attention on PVTv2 [63] and Twins [10].
>
> **Q4.** Why was the ImageNet model not trained from scratch?
>
> **A4.** We follow the standard training settings in binarization literature [40,73]. It has been well explored that on large-scale dataset, fine-tuning from the pre-trained model helps to reduce the performance drop due to the gradient approximation for the non-differentiable $\rm{sign}$ function as explained in L167-169.
>
> To demonstrate the effect of training from scratch, we apply our EcoFormer to PVTv2-B0 as well as PVTv2-B1. We follow the experimental settings mentioned in L253-267 (L252-266 in the previous version) except that we train the model from scratch with 300 epochs. The initial learning rate is set to $2.5 \times 10^{-4}$. From Table IV, our method achieves comparable performance while significantly reducing the computational complexity and energy consumption. The accuracy drop from discretization can be mitigated by more advanced optimization methods as discussed in L126-129. We have put the results in Section E of the supplementary file.
>
> Table IV. Performance comparisons of different methods on ImageNet-1K. All the models are trained from scratch. The number of multiplications, additions, and energy consumption are calculated based on an image resolution of 224 × 224.
>
> |   Model  | Method | #Mul. (B) | #Add. (B) | Energy (B pJ) | Top-1 Acc. (%) |
> |:--------:|:------:|:---------:|:---------:|:-------------:|:--------------:|
> | PVTv2-B0 |   MSA  |    2.02   |    1.99   |      9.3      |      69.72     |
> |          |  Ours  |  **0.54** |  **0.56** |    **2.5**    |    **68.70**    |
> | PVTv2-B1 |   MSA  |    5.02   |    5.00   |      23.1     |      78.34     |
> |          |  Ours  |  **2.03** |  **2.09** |    **9.4**    |    **77.49**   |

---

> > ### Comment · Reviewer_M9L7 · 2022-08-08
> > **Thank you for the response**
> >
> > Thank you for the clarifications. I am satisfied with the responses and maintain my original score.

---

> > > ### Author Response · Authors · 2022-08-09
> > > **Thanks for your feedback**
> > >
> > > Thanks for your feedback and suggestions! We are happy to address your questions and appreciate the valuable comments.

---

> ### Author Response · Authors · 2022-08-02
> **Response to Reviewer M9L7 (Part 2)**
>
> **Q5.** More discussions with the related methods [E][F][G].
>
> **A5.** Thanks for your suggestions. We have included the following discussions in the related work.
>
> *Reformer [30], **SMYRF [E], Fast Transformers [F] and LHA [G]** restrict the attention to the most similar token pairs via hashing and reduce the computational complexity to ${\mathcal O}(N \log N)$. In contrast, our EcoFormer learns kernel-based hash functions using attention scores to map the queries and keys into compact similarity-preserving binary codes in Hamming space, which is energy-efficient and in linear complexity ${\mathcal O}(N)$. With the low-dimensional binary queries and keys, our EcoFormer is able to replace most of the multiplications with simple accumulations.*
>
> **Reference**
>
> [A] Long Range Arena: A Benchmark for Efficient Transformers. ICLR 2020.
>
> [B] Long-short transformer: Efficient transformers for language and vision. NeurIPS 2021.
>
> [C] Nyströmformer: A Nyström-Based Algorithm for Approximating Self-Attention. AAAI 2021.
>
> [E] SMYRF: Efficient Attention using Asymmetric Clustering. NeurIPS 2020.
>
> [F] Fast Transformers with Clustered Attention. NeurIPS 2020.
>
> [G] Sparse Attention with Learning to Hash. ICLR 2022.

---

### Official Review · Reviewer_THJi · 2022-07-12

**Rating:** 5
**Confidence:** 3
**Soundness:** 3 good
**Presentation:** 3 good
**Contribution:** 2 fair

**Summary:**

This paper proposes an efficient attention mechanism with a linear complexity with respect to the context length and only requires bitwise operations. The resulting model is called EcoFormer. Compared to existing kernel-based linear attention approaches, EcoFormer applies the kernelized hash functions to directly map queries and keys into binary codes to further reduce the computational complexity using floating-point values.

**Questions:**

I would like the authors to respond to the main weaknesses mentioned above. In addition, I have a question about how the proposed EcoFormer compares to other non-kernel-based efficient attention schemes (e.g., [2]).

[2] Combiner: Full Attention Transformer with Sparse Computation Cost, NeurIPS'21

**Strengths And Weaknesses:**

Strengths: This idea of applying kernelized hash functions to existing multi-headed self-attention (MHSA) to achieve linear complexity and eliminate the requirement for floating-point arithmetic operations is very interesting. I believe this proposal helps to further minimize the computational cost of MHSA and is considered as a new contribution to the field. The paper is well written and provides a good discussion of the background and related work. The results are also encouraging, as shown in Table 2, where EcoFormer is able to achieve similar quality (i.e., accuracy) to the ordinary MHSA with significantly fewer multiplication and addition operations. Overall, I like the proposed idea and enjoy reading the paper, but cannot recommend accepting the paper because of the following weaknesses.

Weaknesses: My main concern with this paper is whether the savings claimed in the paper can be achieved with existing hardware backends. The savings in terms of computational cost, energy consumption, and area are purely theoretical and are not supported by any actual demonstration.
1) Performance: The actual performance of the proposed kernelized hashing attention is not reported in wall clock time and is not compared with the ordinary MHSA as well as the kernel-based linear attention methods. Many previous works on efficient attention have demonstrated a significant performance improvement compared the ordinary MHSA when the context length is long. For EcoFormer, I hope to understand more about the actual implementation of the proposed attention mechanism. Specifically, I would like to know if the proposed kernelized hashing attention runs faster, slower, or comparable to kernel-based linear attention methods on GPUs or TPUs.
2) Energy: I am not entirely clear about how the energy consumption is calculated. But I think the authors only calculated the energy consumption of arthemitch operations and ignored the energy consumption of DRAM accesses. However, as pointed out by the TPU-v4 paper [1], the energy consumption of DRAM access dominates the energy consumption of the TPU accelerator chip because DRAM access is over 1000 times more expensive than half-precision multiplication. (Please see Table 2 of the paper). Therefore, I think the authors should claim savings in energy consumption by taking the DRAM access energy into account.
3) Area: I am not convinced by the area saving reported in the paper as you cannot simply calculate the area of the accelerator design by multiplying the area of the circuits with the number of operations. The size of the processing element array/compute engine is determined by the memory bandwidth and the data flow of the accelerator to ensure that the accelerator is not constrained by memory or computation in most cases. I think it makes much more sense to claim better performance on a specific hardware backend rather than area savings.

[1] Ten Lessons From Three Generations Shaped Google’s TPUv4i, ISCA'21

---

> ### Author Response · Authors · 2022-08-02
> **Response to Reviewer THJi (Part 1)**
>
> Thanks for your constructive comments.
>
> **Q1.** I would like to know if the proposed kernelized hashing attention runs faster, slower, or comparable to kernel-based linear attention methods on GPUs or TPUs.
>
> **A1.** Our proposed EcoFormer runs faster than the kernel-based linear attention on a GPU. To demonstrate this, we measure the throughput of different methods on a single NVIDIA RTX 3090 GPU. We compare our EcoFormer with the standard multi-head self-attention (MSA) and kernel-based linear attention (KLA) [9]. From Table I, KLA shows higher throughput than MSA, while our EcoFormer achieves even faster throughput than KLA, thanks to the reduced feature dimensions ($b$ vs. $D_p$) of queries and keys (Line 190). With efficient accumulation implementation, the throughput of our EcoFormer can be further improved, which will be explored in the future. We have included these results and the corresponding discussions in Section D of the supplementary material.
>
> Table I. Throughput of different methods on ImageNet-1K. MSA denotes the standard multi-head self-attention and KLA represents the kernel-based linear attention. The throughput is measured with a mini-batch size of 32 and an image resolution of 224×224 on a single NVIDIA RTX 3090 GPU.
>
> | Model       | Method | Test Throughput (images/s) |
> | ----------- | ------ | :--------------------------: |
> | PVTv2-B0    | MSA    | 850  |
> |             | KLA [9]    | 1166  |
> |             | **Ours**   | **1379** |
> | PVTv2-B1    | MSA    | 621 |
> |             | KLA    | 769 |
> |             | **Ours**   | **874** |
> | PVTv2-B2    | MSA    | 404 |
> |             | KLA    | 444 |
> |             | **Ours**   | **483** |
> | Twins-SVT-S | MSA    | 426 |
> |             | KLA    | 489 |
> |             | **Ours**   | **576** |
>
> **Q2.** I am not entirely clear about how the energy consumption is calculated. But I think the authors only calculated the energy consumption of arithmetic operations and ignored the energy consumption of DRAM accesses.
>
> **A2.** We would like to clarify that our EcoFormer only targets at **on-chip** efficiency as mentioned in L16-19, L70-71 and L205-207. Therefore, we only calculate the theoretical energy consumption of arithmetic operations following [54]. The energy consumption of **off-chip** DRAM access is dependent on many system-level factors, such as data reusing, partition and scheduling, which is beyond the scope of the manuscript.
>
> To demonstrate the actual latency and energy consumption with DRAM access taken into consideration, we also test different methods on a simulator of BitFusion [A], a bit-flexible microarchitecture synthesized in 45 nm technology. All our experiments on BitFusion, including our EcoFormer and the other efficient attention mechanisms, use an attention layer with an embedding dimension of 32. We use the sequence length of 3,136 from the first stage of PVTv2 [63]. From Table II, our EcoFormer shows much lower latency and energy than the other efficient attention methods, which further verifies the advantage of our EcoFormer. We have included the results and corresponding descriptions in Section 5.3 of the revision.
>
> Table II. Latency and energy comparisons with different attention methods. We measure the latency and energy of an attention layer with a mini-batch size of 16, a sequence length of 3,136, and an embedding dimension of 32 on a BitFusion [A] simulator.
> |    Method   | Latency (ms)  | Energy (pJ) |
> |:-----------:|:-------------:|:-----------:|
> | Transformer |     0.0036    |   85,692.18  |
> |  Performer [9]  |     0.0019    |   41,113.64  |
> |  Linformer [61]  |     0.0018    |   45,770.61  |
> |   Reformer [30]  |     0.0024    |   57,305.47  |
> |  **EcoFormer**  |     **0.0010**    |   **24,990.75**  |
>
> **Q3.** I am not convinced by the area saving reported in the paper as you cannot simply calculate the area of the accelerator design by multiplying the area of the circuits with the number of operations. I think it makes much more sense to claim better performance on a specific hardware backend rather than area savings.
>
> **A3.** We agree. The on-chip area cost depends on many factors, such as area reusing, memory bandwidth, the data flow of the accelerator, etc. Therefore, we remove the on-chip area cost in our revised manuscript. As mentioned in A2, we instead show the latency and energy on a simulator of BitFusion [A]. From Table II, our EcoFormer significantly saves energy cost and accelerates the inference speed.

---

> > ### Comment · Reviewer_THJi · 2022-08-08
> > **Post-rebuttal**
> >
> > Thank the authors for their response. I agree with the authors that bit fusion can be a potential implementation of the proposed scheme. Moreover, the proposed scheme seems to provide significant energy savings even when considering the access energy of off-chip memory. I will raise my rating to borderline accept.

---

> > > ### Author Response · Authors · 2022-08-09
> > > **Response to post-rebuttal**
> > >
> > > Thanks for your feedback and suggestions! We feel glad to address your questions and appreciate the constructive reviews for improving our work.

---

> ### Author Response · Authors · 2022-08-02
> **Response to Reviewer THJi (Part 2)**
>
> **Q4.** Comparisons with other non-kernel-based efficient attention schemes (e.g., [B]).
>
> **A4.** We have compared two non-kernel-based efficient attention methods (Linformer [61] and Reformer [30]) in Table 6 (Table 5 in the previous version). From the table, our EcoFormer saves much more additions, multiplications, and energy consumption. Besides, we also compare our EcoFormer with Combiner [B] on two tasks, **Text** and **Retrieval** from the Long Range Arena (LRA) benchmark dataset [56]. From Table III, our EcoFormer achieves comparable performance to the other methods while significantly reducing the computational complexity and energy cost. For example, on **Text**, compared with standard Transformer, our method saves around 94.6% multiplications and 93.7% additions as well as 94.5% energy consumption, which is more efficient than the existing efficient attention mechanisms. These results further justify the effectiveness of our EcoFormer under long-context scenarios.
>
> Table III. Performance comparisons of different methods on Long Range Arena (LRA). We report the classification accuracy (%) for **Text** as well as **Retrieval** and the average accuracy across two tasks. Bi-EcoFormer denotes that we use binary quantization [26] instead of our proposed hash functions to obtain binarized queries and keys relying on EcoFormer. $^{*}$ denotes that we obtain the results from the original paper.
>
> |        Model       | #Mul. (B) | #Add. (B) | Energy (B pJ) | Text (4K) | Retrieval (4K) |  Average  |
> |:------------------:|:---------:|:---------:|:-------------:|:---------:|:--------------:|:---------:|
> |      Transformer      |    4.63   |    4.57   |     21.25     |   64.87   |      79.62     |   72.25   |
> |      Performer [9]    |    0.83   |    0.84   |      3.83     |   64.82   |      79.08     |   71.95   |
> |      Linformer [61]     |    0.81   |    0.81   |      3.74     |   57.03   |      78.11     |   67.57   |
> |      Reformer [30]      |    0.54   |    0.54   |      2.49     |   65.19   |      79.46     |   72.33   |
> | Combiner$^{*}$ [B] |    0.51   |    0.51   |      2.34     |   64.36   |      56.10     |   60.23   |
> |  BiQuant-EcoFormer |    0.39   |    0.67   |      2.03     |   64.68   |      75.91     |   70.30   |
> |    **EcoFormer**   |  **0.25** |  **0.29** |    **1.17**   | 64.79 |    78.67   | 71.73 |
>
> **Reference**
>
> [A] Bit fusion: Bit-level dynamically composable architecture for accelerating deep neural network. ISCA 2018.
>
> [B] Combiner: Full Attention Transformer with Sparse Computation Cost. NeurIPS 2021.

---

> ### Author Response · Authors · 2022-08-07
> **Request for discussion**
>
> Dear Reviewer THJi
>
> We sincerely thank you again for your great efforts in reviewing this paper. We have addressed your major concerns regarding the actual demonstration on existing hardware backends (GPU and BitFusion). Please don’t hesitate to let us know if there are still some concerns/questions.
>
> Best regards,
>
> Authors of #231

---

### Author Response · Authors · 2022-08-02
**General Response**

We sincerely thank all reviewers for their valuable comments.

## Novelty
All reviewers recognize the novelty of our method

* "*This idea of applying kernelized hash functions … is very interesting. I believe this proposal helps to further minimize the computational cost of MHSA and is considered as a new contribution to the field.*" (Reviewer THJi)
* "*The idea of using learnable kernelized hashing to preserve similarity relations of binary codes is novel.*" (Reviewer M9L7)
* "*The motivation of this work is promising and binarizing the Transformers through kernelized hashing functions in a self-supervised way is novel.*" (Reviewer BP1F)

## Promising Results
All reviewers agree that

*  "*The results are also encouraging.*" (Reviewer THJi)
*  "*The results are encouraging and demonstrate the effectiveness of the proposed methods on the vision tasks.*" (Reviewer M9L7)
*  "*Experimental results on ImageNet show high efficiency compared with the baselines.*" (Reviewer BP1F)

## Summary of changes
We have revised our submission and summarized our updates as follows:

1. We have provided more throughput results on a GPU. (Reviewer THJi)
2. We have removed the area cost and shown the actual demonstration of different attention mechanisms in terms of energy and latency on BitFusion in the revision. (Reviewer THJi)
3. We have provided more comparisons with the non-kernel-based efficient attention scheme. (Reviewer THJi)
4. We have conducted more experiments in terms of long-context scenarios on the Long-Range Arena benchmark dataset. (Reviewer M9L7)
5. We have provided more details on how to calculate the number of additions and multiplications. (Reviewers M9L7 and BP1F)
6. We have provided more discussions on LSH-based efficient attention methods. (Reviewer M9L7)

---

### Meta-Review · Area_Chair_2mPv · 2022-08-31

**Recommendation:** Accept
**Confidence:** Less certain

**Metareview:**

The paper after rebuttal addresses several of the limitations (mainly lacking positioning in the rich existing litterature) of the first submission. The strength of the paper resides in a holistic approach to the ("yet another") efficient attention mechanism, evaluating and discussing trade-offs between accuracy, compute, energy, and silicon area use.
The main limitations are the limited novelty, and the rather thin experimental validation: no SOTA baselines on ImageNet, LRA not being very correlated to real tasks accuracy.
Overall, I recommend for the paper to be accepted based on its technical merit.

**Award:**

No

---

### Decision · Program_Chairs · 2022-09-14

Accept